# OpenReview forum: "MoCa: Cognitive Scaffolding for Language Models in Causal and Moral Judgment Tasks"
_ICLR.cc/2023/Conference — Submitted to ICLR 2023_

### Official Review · Reviewer_XfnA · 2022-10-15

**Confidence:** 4
**Correctness:** 2
**Technical Novelty And Significance:** 2
**Empirical Novelty And Significance:** 2
**Recommendation:** 3

**Clarity, Quality, Novelty And Reproducibility:**

I've outlined issues with clarity above, which I think are non-trivial, and also affect the reproducibility. I think the quality of the paper can be improved by addressing the various concerns I've raised about interpretation of the results as well as validity of the claims about contribution of insights from cognitive science.

**Strength And Weaknesses:**

Strengths: I'm happy to see engagement with cognitive science literature, and this paper deals with the important topic of models' ability to do humanlike causal and moral reasoning. The experiments are interesting creative.

Weaknesses:

There are issues with methodological clarity, and I'm skeptical about interpretations of a number of the results.

For R2, it seems likely that simplifying to the abstract stories creates simpler lexical cues that the models are able to latch onto to show the bump in performance that is observed. There may also be biases in terms of frequency of mentions, e.g. "A" is mentioned more times in Fig 2a than "B" is. Additionally, it seems worth considering the possibility that humans actually wouldn't show the same patterns with these abstract versions of the stories as with the original stories, since these seem like potentially less intuitive inputs to reason about.

For R3, the authors initially describe the tests as zero-shot, but based on the subsequent description and Appendix examples, it seems what they are doing is few-shot. So this is an issue with methodological clarity, and it makes me concerned about my interpretation of other experiments as well -- as far as I know, the other experiments are truly zero-shot, but I'm not sure if this is actually correct. If that interpretation is correct, however, then the next concern: why is R3 the only experiment where the model is given examples in the prompt? My immediate concern is that the models could identify simple cues in the prompt examples that predict the factor tag (e.g. "both" --> conjunctive). The authors acknowledge something like this concern, and argue that because their prompt examples are higher-level than the final item the model needs to respond to, this concern can be dismissed. However, this really does not exclude the possibility that there are overlapping cues between the prompt examples and the real examples. Additionally, given that R3 is (I think) the only few-shot experiment, and is also the only experiment in which the models show notably higher performance than any of the other experiments, this raises further suspicion that the strong performance is attributable to simpler solutions based on the few-shot examples.

As for R4, I'm having trouble understanding what we stand to gain/learn from this experiment, and how we should interpret improvements relative to R1/R2, since it seems that the R4 is simply adding less accurate factor labels to what is otherwise the R2 setup. I'm (again) not completely clear on the method here, but as far as I can tell R4 is just changing the list of labels prepended to the abstract story from R2, and making them the GPT-3 predicted labels, which are decent but worse. What do we stand to learn from this? Why is it useful to check how the models perform if the labels become the GPT-3 labels? The results table only compares to R1, so I'm having to scroll back and forth to try to check the (seemingly more direct) comparison with R2 -- it looks like it's often worse, but sometimes on par/better.  So is the conclusion that if we do a noisier version of R2, performance is often worse but sometimes better? Why would that be? Something else that is unclear: the description of the "Expert Reasoning Engine" comes out of the blue and is not enough for me to be clear what was done (and is also confusing because it uses the phrase "one can imagine" multiple times, suggesting this is merely a hypothetical, but then it turns out to be in the reported results). Did the authors train a separate classifier mapping between GPT-3's noisy labels and the "Yes/No" labels? Is this no longer a cloze task but a classification? There really needs to be a lot more methodological clarity here.

Finally, zooming out a bit more: the authors are claiming that cognitive science-based scaffolding is helpful for the models' performance, but I'm not sure in what way this is shown. It seems like it's mostly the labeling of factors that comes from the cognitive science literature, but it's not clear that listing those factors along with the story (which I think, but am not sure, is what's being done) is actually making much of a contribution to models being able to make humanlike judgments on the stories. The other candidate for "cognitive scaffolding" is the translation of the stories into abstract versions -- this does seem to help the models, but it's not clear that this is a contribution from cognitive science, and I also had concerns about lexical shortcuts as outlined above.



**Summary Of The Paper:**

This paper examines behaviors of large language models in causal and moral judgment tasks drawn from cognitive science literature studying human judgments in these domains. They first test on simple cloze tasks using stories and prompts that human participants responded to, and identify model preferences for continuations of "Yes" or "No". They find low (near-chance) levels of alignment with human judgments. However, when they simplify to giving the LMs abstract descriptions capturing high-level properties of the stories, alignment improves a bit. They then do a factor inference task, where the models are supposed to complete a prompt with a factor label from the cognitive science literature. Lastly they combine the latter two experiments, getting tags from GPT-3 and using these to along with an abstract prompt. This last experiment shows a small amount of improvement in the largest GPT-3 model.

**Summary Of The Review:**

I am happy to see this engagement with cognitive science, and I think this is an interesting and useful topic -- but I have numerous non-trivial concerns about methodological clarity and validity of conclusions, and I'm skeptical that the paper has given a strong demonstration of usefulness of insights from cognitive science, as it claims. I think these concerns should be addressed before publication, but I am optimistic that the paper can be much improved and make an interesting contribution.

---

> ### Author Response · Authors · 2022-11-17
> **Response (1)**
>
> Thank you for your thoughtful questions and comments. We really appreciate the time you took to give the feedback with the goal of helping us improve the clarity of the paper. We want to highlight that our work uses cognitive science to bring **latent structure** to plain text stories and demonstrate how we can use these latent structures to produce more human-aligned results for LLMs. We believe this is a key advantage to our dataset – the writing of the scenarios is **tightly controlled and carefully designed by cognitive scientists**. Additionally, the participants' way of responding to these scenarios was restricted to give ways to scientific, quantitative evaluation.
>
> We will address your concern point by point.
>
> > There may also be biases in terms of frequency of mentions, e.g. "A" is mentioned more times in Fig 2a than "B" is.
>
> We agree that the frequency of person mentions can be different, but our label distribution is roughly balanced across three labels (Yes/No/Ambiguous). Simply biasing the model to generate one label will not improve performance overall.
>
> | Dataset | # Stories | Yes (p > 0.6) | No (p < 0.4) | Ambiguous | # words per story | # words per translated story |
> |---------|----------:|:-------------:|:------------:|:---------:|:-----------------:|:----------------------------:|
> | Causal  |       144 |       48      |      50      |     46    |        162        |             82.9             |
> | Moral   |        62 |       23      |      10      |     29    |        72.5       |             53.5             |
> | Total   |       206 |       71      |      60      |     75    |        135        |             74.1             |
>
> > Additionally, it seems worth considering the possibility that humans actually wouldn't show the same patterns with these abstract versions of the stories as with the original stories since these seem like potentially less intuitive inputs to reason about.
>
> Thank you for raising this point. We want to verify if this is indeed true and conduct another experiment with humans. We use the same setup as we did for the human label collection for the original stories. We collect 25 responses per translated story to calculate P(Yes) and use it the same way as if the probability was generated from a language model in our evaluation pipeline.
>
> We notice that it is harder for people to conduct causal reasoning on these abstract stories than real stories (crowdsourced workers also take less time due to inattention and anecdotally told us these stories are less enjoyable than real stories). This additional study perhaps shows that human reasoning is more nuanced, and although our proposed translation helps GPT-3, it might not always elicit the same responses from humans. We will include this in the paper along with our discussion. We believe this is exciting to share with the rest of the community, particularly on the divergence between LLM and Human Cognition.
>
> | Causal      | Accuracy | AUC    | MAE    | CE     |
> |-------------|----------|--------|--------|--------|
> | InstructGPT | 0.2586   | 0.6333 | 0.2910 | 0.6016 |
>
> | Moral       | Accuracy | AUC    | MAE    | CE     |
> |-------------|----------|--------|--------|--------|
> | InstructGPT | 0.3276   | 0.6214 | 0.2741 | 0.5842 |
>
> > For R3, the authors initially describe the tests as zero-shot, but based on the subsequent description and Appendix examples, it seems what they are doing is a few-shot.
>
> You are correct – thank you for catching this. We have updated our terminology to reflect the few-shot setup of the task in the paper. The rest of the experiments are indeed zero-shot. We have provided the code in the supplementary material for additional verification.

---

> > ### Author Response · Authors · 2022-11-17
> > **Response (2)**
> >
> > > why is R3 the only experiment where the model is given examples in the prompt? …this really does not exclude the possibility that there are overlapping cues between the prompt examples and the real examples. Additionally, given that R3 is (I think) the only few-shot experiment, and is also the only experiment in which the models show notably higher performance than any of the other experiments, this raises further suspicion that the strong performance is attributable to simpler solutions based on the few-shot examples.
> >
> > We use a few-shot setup for R3 because we believe we can write clear examples to “teach” LLMs how to identify relevant factors such as (causal structure, agent awareness, etc.). However, we do not have a good way to create few-shot examples for R1 manually. Indeed, we could randomly sample K items from the dataset and use them as few-shot examples; but this defeats the purpose of using our dataset as an evaluation dataset to measure zero-shot alignment on causal and moral judgment.
> >
> > We are not particularly concerned that R3 performance is much higher than the rest of the tasks. On a high level, LLMs have demonstrated a remarkable ability to assign the right labels for tasks such as sentiment classification and document labeling. Many of the tasks in R3 (causal structure prediction, agent awareness prediction) are not so different from these tasks. It also will not create unfair comparisons between models and tasks. R3 is part of our scaffolding step, not the main task that we focus on. Our main goal is to help LLMs align with human judgments in a zero-shot setting.
> >
> > > As for R4, I'm having trouble understanding what we stand to gain/learn from this experiment…how we should interpret improvements relative to R1/R2. Why is it useful to check how the models perform if the labels become the GPT-3 labels?
> >
> > We have rewritten the introduction in R4 to make it more clear. In R1, we introduce the task and show a baseline model performance. In R2, we use gold labels for relevant factors in each story from human annotation. In R3, we use LLMs to infer these annotations and found that through a few-shot setup, LLMs can infer these annotations accurately. In R4, we put everything together without relying on human gold labels; this partly tests how well GPT3 could work by itself in a more generalizable setting (without human labels), for example, on other stories for which we don't have human labels on the latent structures.
> >
> > > The results table only compares to R1, so I'm having to scroll back and forth to try to check the (seemingly more direct) comparison with R2 -- it looks like it's often worse, but sometimes on par/better. So is the conclusion that if we do a noisier version of R2, performance is often worse but sometimes better? Why would that be?
> >
> > We use R4 as an end-to-end pipeline that we propose to improve R1. It’s not comparable to R2, and we expect it to be worse than R2 because we are not using human gold labels. To provide more context, these stories are written by cognitive scientists to investigate whether a particular factor will influence human judgment (they answer research questions like: “If someone is unaware of their action, will people still judge them to have caused the outcome?”).
> >
> > > "Expert Reasoning Engine" comes out of the blue and is not enough for me to be clear what was done (and is also confusing because it uses the phrase "one can imagine" multiple times, suggesting this is merely a hypothetical, but then it turns out to be in the reported results). Did the authors train a separate classifier mapping between GPT-3's noisy labels and the "Yes/No" labels? Is this no longer a cloze task but a classification?
> >
> > Your interpretation is correct. ERE is a separately trained classifier that maps GPT-3 labels to Yes/No (not trained on the gold labels, but on a separate set of labels we transcribed from the original papers). We are using ERE to demonstrate why using LLMs to infer **latent structure** in the text is beneficial, as it creates new ways of increasing alignment (turning a free-text generation problem into a binary classification problem).

---

> > > ### Author Response · Authors · 2022-11-17
> > > **Response (3)**
> > >
> > > > It seems like it's mostly the labeling of factors that comes from the cognitive science literature, but it's not clear that listing those factors along with the story (which I think, but am not sure, is what's being done) is actually making much of a contribution to models being able to make humanlike judgments on the stories. The other candidate for "cognitive scaffolding" is the translation of the stories into abstract versions -- this does seem to help the models, but it's not clear that this is a contribution from cognitive science.
> > >
> > > Our work is influenced by cognitive science in two ways: 1). The dataset has been transcribed from 24 cognitive science papers; 2). The relevant factors we discuss in the paper have been proposed, systematically studied, and examined by cognitive scientists. These factors bring structure to otherwise plain text stories. Our approach uses these latent structures in the story to make advance in aligning model responses with human intuitions. We argue that it is difficult to come up with an alternative structure without bringing in inspiration from three decades of scientific research in cognitive science. In conclusion, we believe cognitive science laid the foundation for our work.
> > >
> > > > I think these concerns should be addressed before publication, but I am optimistic that the paper can be much improved and make an interesting contribution.
> > >
> > > We really appreciate the methodology and clarity issues you brought up in your review. It has shaped how we think about our papers and communicate our ideas. We don't believe there is any core methodological flaw in the paper, as we are focusing on 1). Summarizing and bringing 30 years of Cognitive Science research into one paper (see Table 1 and Appendix A.1 A.2). This provides unique interdisciplinary value; 2). Demonstrating how these tightly controlled stories can help measure AI/human alignment in this particular setting; 3). Showing that the cognitive science framework can inspire new ideas in causal and moral judgment tasks (thought-as-text translation, ERE). We are hoping this paper can be a good reference to other AI researchers working in similar areas and influence future dataset collection efforts and design.
> > >
> > > We have updated the text to reflect your comments. We are hoping these responses have addressed your concerns. We are happy to engage in additional discussions for further clarification!

---

### Official Review · Reviewer_ikDj · 2022-10-24

**Confidence:** 4
**Correctness:** 2
**Technical Novelty And Significance:** 3
**Empirical Novelty And Significance:** 3
**Recommendation:** 3

**Clarity, Quality, Novelty And Reproducibility:**

Clarity:
- There are a number of typos in the dataset, e.g., "colleccted", "theoriezed", "severval", "inter-rator"; also in the "Dataset" paragraph of 3.1, there seems to be a partial sentence cut off.
- Citations for COPA and SuperGLUE appear to be transposed

Quality:
- The experiments are thorough.
- The dataset itself could be valuable, especially for cogsci researchers, but there should be more details provided about its collection and the distribution of annotations in each story. The framing of questions and stories could also affect the moral judgments in a way that results in biases in human decision-making.

Novelty:
- The dataset itself appears to be new, and relatively thorough (reaching back to decades of cogsci research to acquire more high-quality scenarios).

**Details Of Ethics Concerns:**

Please see above (summary of the review) for a description of several assumptions that I feel this paper is making about judgments of causality and morality.

I worry that any systems which we try to "align" to normative human judgments (including judgments about causality and morality) have no reason to behave ethically, fairly, etc. (partially because we know, e.g., from large datasets of text and images, that "normative" data replicates biases).

**Strength And Weaknesses:**

Strengths:
- The description of the task and crowdsourcing (especially expert judgment) annotation is quite thorough.
- Experiments are easy to understand.
- Evaluation of the LLM by disentangling text complexity with reasoning is interesting (Section 4.2).

Weaknesses:
- There are many missing details about the crowdworkers involved in the judgment tasks. Who was recruited to participate? Where were they located?
- There is also missing detail about the dataset itself. Perhaps the most interesting part of this dataset might be the actual distributions (between yes/no) of user annotations on the stories, but this is not provided in the paper or materials as far as I can tell. Instead, it appears that stories are bucketed into "yes", "no", and "ambiguous" buckets (when agreement is too low).
- There is missing discussion on the motivations and framing of these tasks at a meta-level. "when something bad happened, we naturally ask: who did what, and why?", "people's moral knowledge helps them tell apart good from bad" -- these claims are from a particular philosophical perspective, e.g. where "good" and "bad" are standalone values that can be placed on an action or event in absence of enough context.
- Similarly, framing the task of causality as placing blame on an agent rather than on the structural context is a specific choice. I understand this study is coming from the perspective that cogsci has created on causality, but this framing shapes event understanding in a particular way (i.e., that structural context, e.g., that Billy/Suzy's computer has this particular flaw, is not the focus of blame). I also understand that the goal is to study how predictions correlate with human judgments, but it also seems that the questions asked of annotators ("Did Billy cause the emails to be deleted?"), rather than asking open-ended questions, may result in a bias towards agents becoming the sole focus of blame in LLMs (if we want to assume that LLMs will or should adopt the moral judgments of humans as studied in tasks like these).
- The metrics themselves are not well-motivated. Accuracy with discretized categories of judgments (yes, no, ambiguous) seems like an arbitrary choice. Did the authors consider directly evaluating the distribution similarity between the LLM's prediction and the distributions among 25 human raters?
- It appears there isn't human evaluation for the generated T3 stories, which would make the comparison more accurate in Table 3.
- I don't think the claim "we ... only test if LLMs can have any implicit or innate ability to combine these factors and produce the same judgment a human would" is accurate. It is still possible that the authors' choice in generating the T3 versions of stories influences the model's predictions (e.g., with a spurious bias towards "yes" across the stories) simply in how they were worded, so although the details of stories are removed, the surface form itself is still not being controlled for.
- I'd be interested in seeing "error analysis" of the predictions in Table 4, as well as precision/recall of factor prediction rather than plain accuracy.
- There is an implication in Section 5 that "aligning" our LLMs to perform well in these causal/moral judgment tasks will result in better text generation in general (please correct if I am wrong about this implication), yet there isn't really substantial evidence of these LLMs (1) being "misaligned" with human judgments in general, free-form text generation, or that (2) "aligning" them with the proposed method will improve "alignment" in free-form text generation.

**Summary Of The Paper:**

This paper studies how well-correlated human judgments and LLM predictions are for causal and moral reasoning tasks. They collect a dataset of human judgments for a number of text problems used in cognitive science literature for probing human judgments, including expert judgments for relevant factors in the judgment process (e.g., norms, inevitability). They perform a few experiments that find that (1) correlation between human and LLM predictions is low out-of-the-box, but that (2) explicitly providing factors only, or combining the two methods, can yield better performance.

**Summary Of The Review:**

This paper is thorough, and the paper is clearly written. However, I am a bit wary of underlying assumptions made in the design of the paper, which makes me lean towards rejection:

- That moral and causal judgments can be bucketed into "yes", "no", and "ambiguous".
- That reasoning about causality is equivalent to answering the question of which agent is to blame under a particular structural context, rather than placing blame on the structural context of the event itself.
- That we should prioritize "aligning" LLMs with some measure (in this paper, majority vote) of human judgments in terms of morality and causality as it's studied from the perspective of cogsci researchers, even though human judgments, reasoning, and perspectives about morality and causality vary significantly across personal experience, cultures, etc. In particular, this paper is focusing on a set of crowdworkers to provide judgments (while providing no details about how those workers were selected).

---

> ### Author Response · Authors · 2022-11-16
> **Response (1)**
>
> Thank you for your thoughtful questions and comments. We apologize that it took longer for us to respond to your review. We ran additional human studies and will discuss the results shortly. Our task focuses on measuring alignment in a restrictive cognitive science setup, different from investigating causal and moral reasoning as a whole. In our study: of causal/moral judgments, we want to quantitatively investigate and understand people and agents’ abilities and biases when conducting causal/moral-based social reasoning (therefore, the questions almost exclusively focus on actors/characters in the story). Different types of setups will have other implications. Still, in this work, we want to take the first step by leveraging three decades of research in cognitive science and experimental philosophy.
>
> > There are many missing details about the crowdworkers involved in the judgment tasks. Who was recruited to participate? Where were they located?
>
> Thank you for raising this question. We apologize for not making it clear what is included in our appendix. Here is the paragraph in Appendix A.10 describing our data collection process:
>
> We recruit participants from Prolific to give responses to our stories. For each story, we elicit a yes/no answer from a single participant. We pay wages equivalent to $12/hr and we only recruit participants from the US and UK, with English as their first language. Each participant is presented with 5-6 stories either from the causal dataset or the moral dataset. We present the story and question, and then ask them to choose yes or no. The interface is designed so that the participant is presented with one story at a time. Participants are allowed to change their decisions to any story before final submission. We keep track of how long they spend on each story. We do not collect personal, private information of any sort from the participants except their yes/no responses. All the raw data from our experiment is provided in the supplement.
>
> We also updated the main text to make it clear: “We describe the experiment design and data collection details in Appendix A.10.”
>
> > There is also missing detail about the dataset itself. Perhaps the most interesting part of this dataset might be the actual distributions (between yes/no) of user annotations on the stories…
>
> Thank you for making this suggestion. We have added a new table in the main text that describes the distribution of our bucketed human labels (which we show below as well). We also provide two figures in Appendix Fig A.3 to show the distribution of P(“Yes”) (and P(“No”) can be read as the inverse of the P(“Yes”) graph) [anonymous image link](https://imgur.com/a/gtBdy4W).
>
>
> | Dataset | # Stories | Yes (p > 0.6) | No (p < 0.4) | Ambiguous | # words per story | # words per translated story |
> |---------|----------:|:-------------:|:------------:|:---------:|:-----------------:|:----------------------------:|
> | Causal  |       144 |       48      |      50      |     46    |        162        |             82.9             |
> | Moral   |        62 |       23      |      10      |     29    |        72.5       |             53.5             |
> | Total   |       206 |       71      |      60      |     75    |        135        |             74.1             |
>
> > There is missing discussion on the motivations and framing of these tasks at a meta-level. "when something bad happened, we naturally ask: who did what, and why?", "people's moral knowledge helps them tell apart good from bad" -- these claims are from a particular philosophical perspective, e.g. where "good" and "bad" are standalone values that can be placed on an action or event in absence of enough context.
>
> We apologize for the confusion! We are not promoting a specific philosophical perspective. Instead, we argue that given a context (a story), humans can make causal and moral judgments, a form of social inference on the agents described in the story. We aim to evaluate LLMs' capability to perform these social inferences and be influenced by the same set of latent factors that also influence human judgments.

---

> > ### Author Response · Authors · 2022-11-16
> > **Response (2)**
> >
> > > Similarly, framing the task of causality as placing blame on an agent rather than on the structural context is a specific choice…but this framing shapes event understanding in a particular way (i.e., that structural context, e.g., that Billy/Suzy's computer has this particular flaw, is not the focus of blame)...it also seems that the questions asked of annotators ("Did Billy cause the emails to be deleted?"), rather than asking open-ended questions, may result in a bias towards agents becoming the sole focus of blame in LLMs (if we want to assume that LLMs will or should adopt the moral judgments of humans as studied in tasks like these).
> >
> > Thank you for bringing up this point. Our main focus is on using a principled framework to analyze whether LLMs are also influenced by the factors that systematically influence people’s causal and moral judgments. From this perspective, we think the experiments we conduct in the paper highlight the misalignment between the current LLM system and human cognition. Whether LLMs should mimic human behavior is up for debate – but at least we should allow some work to focus on the measurement of alignment. However, we do acknowledge the limitation of the cognitive science experimental design of this task, where the focus is often on agents (due to the desire to understand people’s social reasoning biases), not on objects (as you pointed out). There are a handful of stories in our dataset where the people are asked to determine if a machine is at fault, but they didn’t make up for a significant portion of our stories.
> >
> > > The metrics themselves are not well-motivated. Accuracy with discretized categories of judgments (yes, no, ambiguous) seems like an arbitrary choice. Did the authors consider directly evaluating the distribution similarity between the LLM's prediction and the distributions among 25 human raters?
> >
> > Thank you for this suggestion! We have included MAE (mean-absolute error) that measures the difference between the probability of human judgment and LLM-generated text token’s probability. To make this more clear, we are also including CE (Cross-Entropy). We updated the table in the text and show them below:
> >
> > |                   | Causal |      |      | Moral |      |      |
> > |-------------------|:------:|:----:|:----:|:-----:|:----:|:----:|
> > | Models            |   AUC  | MAE  |  CE  |  AUC  | MAE  |  CE  |
> > | RoBERTa-large     |  0.50  | 0.50 |  2.6 |  0.50 | 0.50 | 2.65 |
> > | ALBERT-xxlarge    |  0.51  | 0.38 | 1.37 |  0.48 | 0.37 | 1.82 |
> > | Electra-gen-large |  0.53  | 0.44 | 1.94 |  0.49 | 0.46 | 1.59 |
> > | GPT2-XL           |  0.49  | 0.42 | 1.47 |  0.51 | 0.43 | 1.71 |
> > | GPT3-Medium       |  0.47  | 0.33 | 0.74 |  0.43 | 0.41 | 1.03 |
> > | GPT3-Large        |  0.51  | 0.37 | 1.14 |  0.45 | 0.39 | 1.32 |
> > | InstructGPT       |  0.61  | 0.31 | 0.72 |  0.67 | 0.30 | 0.74 |
> >
> > > It appears there isn't human evaluation for the generated T3 stories, which would make the comparison more accurate in Table 3. It is still possible that the authors' choice in generating the T3 versions of stories influences the model's predictions (e.g., with a spurious bias towards "yes" across the stories) simply in how they were worded, so although the details of stories are removed, the surface form itself is still not being controlled for.
> >
> > Thank you for raising this point. We are not biasing the model to generate yes (because our dataset is roughly balanced across yes and no labels). We have evaluated this story with humans. We use the same human experiment setup as we did for the human label collection for the original stories. We collect 25 responses per translated story (with $1200 spent in total for all translated stories) to calculate P(Yes) and use it as if the probability was generated from a language model in our evaluation pipeline.
> >
> > | Causal      | Accuracy | AUC    | MAE    | CE     |
> > |-------------|----------|--------|--------|--------|
> > | InstructGPT | 0.2586   | 0.6333 | 0.2910 | 0.6016 |
> >
> > | Moral       | Accuracy | AUC    | MAE    | CE     |
> > |-------------|----------|--------|--------|--------|
> > | InstructGPT | 0.3276   | 0.6214 | 0.2741 | 0.5842 |
> >
> > We notice that people spend less time making causal judgments in these abstract stories compared to the original stories (crowdsourced workers anecdotally told us these stories are less enjoyable than real stories). This additional study perhaps shows that human reasoning is more nuanced, and although our proposed translation helps GPT-3, it might not always elicit the same responses from humans. We will include this in the paper along with our discussion. We believe this is exciting to share with the rest of the community, particularly on the divergence between LLM and human cognition.

---

> > > ### Author Response · Authors · 2022-11-16
> > > **Response (3)**
> > >
> > > > I'd be interested in seeing "error analysis" of the predictions in Table 4, as well as precision/recall of factor prediction rather than plain accuracy.
> > >
> > > We have provided both precision and recall tables in the appendix Section A.6. We look into the sentences that the model often gets wrong, and we notice that some more ambiguous tasks (such as “personal force,” where we have trouble giving precise definitions and instructions to GPT) often fare poorly compared to tasks where it’s easy to define (such as “norm type”). Here's the recall table:
> > >
> > > | Recall      | Action Omission | Norm Type |  Time  | Agent Awareness | Causal Structure | Event Normality |
> > > |-------------|----------------:|:---------:|:------:|:---------------:|:----------------:|:---------------:|
> > > | GPT3 Size   |         (N=112) |   (N=88)  | (N=67) |      (N=43)     |      (N=98)      |      (N=89)     |
> > > | Medium      |            28.6 |    76.1   |  19.4  |       69.8      |       32.7       |       49.4      |
> > > | Large       |            32.1 |    23.9   |   7.5  |       46.5      |       67.3       |       49.4      |
> > > | InstructGPT |            92.0 |    97.7   |  85.1  |      100.0      |       98.0       |       84.3      |
> > >
> > > | Recall      | Personal Force | Beneficience  | Locus of  Intervention | Causal Role | Counterfactual Evitability |
> > > |-------------|---------------:|:-------------:|:----------------------:|:-----------:|:--------------------------:|
> > > | GPT3 Size   |      N=48      |      N=48     |          N=48          |     N=48    |            N=48            |
> > > | Medium      |           50.0 |      54.2     |          40.0          |     37.5    |            56.2            |
> > > | Large       |           50.0 |      50.0     |          40.0          |     62.5    |            54.2            |
> > > | InstructGPT |           75.0 |      75.0     |          90.0          |     64.6    |            68.8            |
> > >
> > > > There is an implication in Section 5 that "aligning" our LLMs to perform well in these causal/moral judgment tasks will result in better text generation in general (please correct if I am wrong about this implication), yet there isn't really substantial evidence of these LLMs (1) being "misaligned" with human judgments in general, free-form text generation, or that (2) "aligning" them with the proposed method will improve "alignment" in free-form text generation.
> > >
> > > Thank you for raising this point. Section 5 is our ethical statement. Free-form text generation is different from what we are evaluating. We adopt a somewhat restrictive experiment setup inspired by and similar to the cognitive science setups for human participants. We believe this is a key advantage to our dataset – the writing of the scenarios is tightly controlled and carefully designed. Additionally, the participants' way of responding to these scenarios was restricted to give ways to scientific, quantitative evaluation (compared to reading free-form text, which usually is a qualitative evaluation).
> > >
> > > We show that LLM outputs and human judgments are misaligned under a similar experimental setup. Indeed, we can elicit free-form responses from LLMs and bring in crowd-sourced workers to annotate whether these responses are aligned with human responses – but that can introduce its own form of bias into the assessment of alignment.
> > >
> > > > I worry that any systems which we try to "align" to normative human judgments (including judgments about causality and morality) have no reason to behave ethically, fairly, etc. (partially because we know, e.g., from large datasets of text and images, that "normative" data replicates biases).
> > >
> > > Thank you for bringing up this point. We believe checking for alignment is useful, and then we can understand whether alignment happened for the right reasons. People have biases, and these biases are the byproduct of their background, culture, upbringing, and socioeconomic status. Aligning with one particular group of people might lead to misalignment with a different group of people. However, causal and moral judgments make up a non-trivial portion of our everyday social reasoning. If we are hoping to use LLMs in different everyday settings, we want to fully understand the difference between LLMs and humans, which makes alignment research like ours useful and a meaningful contribution to the field.
> > >
> > > > There are a number of typos in the dataset…Citations for COPA and SuperGLUE appear to be transposed
> > >
> > > Thank you for catching these – we have fixed these issues and updated our text in the revised version.
> > >
> > > We really appreciate your long and detailed review. We have learned a great deal by going through your comments, which helped us understand where the paper is not clear and how we didn't communicate some of our ideas across. Your comments are invaluable during our paper revision process, and we truly appreciate them.
> > >
> > > Have we addressed your questions and comments appropriately? We are happy to discuss or answer more questions if they come up.

---

### Official Review · Reviewer_FUkn · 2022-10-28

**Confidence:** 4
**Correctness:** 4
**Technical Novelty And Significance:** 4
**Empirical Novelty And Significance:** 4
**Recommendation:** 8

**Clarity, Quality, Novelty And Reproducibility:**

The clarity of the paper could be improved. This is the main weakness of the paper right now. It's just a lot to take in, and I suspect readers will have trouble skimming it. There is plenty of novelty, and there are enough details for reproducibility.

**Strength And Weaknesses:**

Strengths:
- The problem area is important.
- The proposed methods are intuitively reasonable, and they seem to work well.

Weaknesses:
- This paper has a lot to take in. I found myself needing to jump back and forth between R1, R3, etc.

Questions:
- Could the proposed methods for improving performance on these tasks also be used to improve performance on existing datasets, e.g., ETHICS? This would greatly strengthen the paper.


Minor points:
- Typo: "With the crowd sourced votes, we can make our Since each of the story is designed to verify a scientific"
- Typo: "The inter-rator agreement"

**Summary Of The Paper:**

This paper investigates whether large language models can make causal judgments and reason about moral permissibility in text scenarios. Two new datasets are collected by aggregating stories from 24 cognitive science papers and standardizing the human annotations from the datasets accompanying these papers into ML-ready datasets (additional expert annotation is performed as well). The authors find that LLMs perform poorly on both datasets. To improve performance, they propose three methods: cognitive scaffolding (predicting useful intermediate attributes), thought-as-text translation (abstracting away irrelevant details), and expert reasoning engine (predicting label directly from the intermediate attributes with a shallow network). In experiments, these additions greatly improve performance.

**Summary Of The Review:**

I think this paper could generate good discussion, and the new datasets seem like useful tests for LLMs. I vote to accept.

-----------------------
Update after reading rebuttal and other reviews:

The other reviewers raise some interesting points. In particular, Reviewer XfnA's point about simplifying stories and lexical cues may need to be further addressed. (Note: I'm not convinced that the paper has serious ethical concerns.) Overall, I'm still convinced that this paper is valuable. This is mainly because the only real avenue people have considered until now for increasing performance on ethics / morality benchmarks is using bigger LLMs. While this does work, it's very valuable to explore methods that differentially improve performance on these tasks using domain knowledge. This work accomplishes precisely that, so I think it provides value to the community. Consequently, I'm keeping the same recommendation of acceptance.

---

> ### Author Response · Authors · 2022-11-16
> **Response**
>
> We would like to thank the reviewer for their thoughtful review and positive feedback that our paper can generate good discussions in an important problem area, our datasets are useful for future ML research, and our proposed methods are novel. We really appreciate the feedback you have given and will work to incorporate all your suggestions. We will address your questions below.
>
> > This paper has a lot to take in. I found myself needing to jump back and forth between R1, R3, etc.
> Thank you for raising this concern. We have rephrased the paragraphs to build more connections between each of our experiments.
>
> > Could the proposed methods for improving performance on these tasks also be used to improve performance on existing datasets, e.g., ETHICS? This would greatly strengthen the paper.
>
> Thank you for the great suggestion! The type of ethical dilemma we are trying to resolve is slightly different from the stories presented in ETHICS. We focus on longer, carefully curated stories with more nuanced moral reasoning. ETHICS dataset focuses on clearcut moral scenarios with high agreement among annotators. However, through non-trivial extension, we believe our proposed step-by-step reasoning framework, by analyzing a story through causal roles, beneficience, personal force, etc., can be adapted to evaluate on datasets like ETHICS.
>
> > Typo: "With the crowd sourced votes, we can make our Since each of the story is designed to verify a scientific"
>
> > Typo: "The inter-rator agreement"
>
> Thank you for catching these – we have fixed these issues and updated our text in the revised version.
>
> Overall, we believe our task and dataset can make a significant contribution to the field of moral/causal AI and AI/human alignment research. We are grateful for your appreciation and recognition of our work. It is often difficult to publish in a contentious area like this, where there is not a single right answer. However, we hope by bringing insights from cognitive science and experimental philosophy, as well as summarizing 30 years of work into one single paper, we believe our work stands as a unique contribution to the community at large.
>
> Have we addressed your questions and comments appropriately? We are happy to discuss or answer more questions if they come up.

---

### Decision · Program_Chairs · 2023-01-20

**Decision:**

Reject

**Justification For Why Not Higher Score:**

how the proposed method is inspired by cognitive science needs further justifications.
some technical steps in the method need further clarifications.

**Justification For Why Not Lower Score:**

n/a

**Metareview: Summary, Strengths And Weaknesses:**

This paper investigates whether the language models can make causal judgements and reason about moral permissibility. To this end, the authors collected two datasets by aggregating stories from 24 cognitive papers. By conducting experiments on these two datasets, the authors show that the existing language models do not perform well. Three methods were proposed to improve the performance.

This paper studies an interesting and important problem. The research questions are clear and the answers to these questions provide new knowledge. However, there are a series of concerns raised by the reviewers. For instance,  simplifying to the abstract stories creates simpler lexical cues that the models are able to latch onto.  Framing the task of causality as placing blame on an agent rather than on the structural context is a specific choice that lacks detailed justifications. The method part is clearly described and it is not clear how cognitive science-based scaffolding is helpful to improve performance. Given the concerns, the paper needs a major revision before publication. Although we think the paper is not ready for ICLR in this round, we believe that the paper would be a good one if the concerns can be well addressed.